# Piezo1 links mechanical forces to red blood cell volume

**Stuart M Cahalan[1†], Viktor Lukacs[1†], Sanjeev S Ranade[1], Shu Chien[2,3], Michael Bandell[4], Ardem Patapoutian[1]\***

[1]Department of Molecular and Cellular Neuroscience, Howard Hughes Medical Institute, The Scripps Research Institute, La Jolla, United States; [2]Department of Bioengineering, University of California, San Diego, San Diego, United States; [3]Institute of Engineering in Medicine, University of California, San Diego, San Diego, United States; [4]Genomics Institute of the Novartis Research Foundation, San Diego, United States

**Abstract** Red blood cells (RBCs) experience significant mechanical forces while recirculating, but the consequences of these forces are not fully understood. Recent work has shown that gain-of-function mutations in mechanically activated Piezo1 cation channels are associated with the dehydrating RBC disease xerocytosis, implicating a role of mechanotransduction in RBC volume regulation. However, the mechanisms by which these mutations result in RBC dehydration are unknown. In this study, we show that RBCs exhibit robust calcium entry in response to mechanical stretch and that this entry is dependent on Piezo1 expression. Furthermore, RBCs from blood-cell-specific Piezo1 conditional knockout mice are overhydrated and exhibit increased fragility both in vitro and in vivo. Finally, we show that Yoda1, a chemical activator of Piezo1, causes calcium influx and subsequent dehydration of RBCs via downstream activation of the KCa3.1 Gardos channel, directly implicating Piezo1 signaling in RBC volume control. Therefore, mechanically activated Piezo1 plays an essential role in RBC volume homeostasis.

\*For correspondence: ardem@ scripps.edu

[†]These authors contributed equally to this work

**Competing interests:** The authors declare that no competing interests exist.

## Introduction

Mammalian red blood cells (RBCs) are rather unique in that they lack a nucleus and many organelles and that they traverse the circulatory system several hundred thousand times in their life cycle. RBCs experience significant mechanical forces while recirculating that influence their physiology in many ways, including changes in deformability (*Chien, 1987*), ATP release (*Sprague et al., 2001*), NO release (*Yalcin et al., 2008*), and $Ca^{2+}$ influx (*Larsen et al., 1981*; *Dyrda and et al., 2010*), the latter of which can influence RBC volume. Changes in RBC volume can affect their membrane integrity and ability to travel through capillaries with diameters smaller than the RBCs themselves. The critical importance of RBC volume regulation is demonstrated by several pathologies resulting from either overhydration or dehydration of RBCs (*Gallagher, 2013*). However, the molecular mechanisms by which RBCs sense mechanical forces and the effects of these forces on volume homeostasis have remained unclear.

Recent studies have identified a conserved family of mechanosensitive non-selective cation channels, Piezo1 and Piezo2 (*Coste et al., 2010*, *2012*). Piezo1 responds to a wide array of mechanical forces, including poking, stretching, and shear stress, and is essential for proper vascular development in mice (*Nilius and Honore, 2012*; *Li et al., 2014*; *Ranade et al., 2014*). The potential role of Piezo1 in RBC physiology is most clearly demonstrated by many gain-of-function mutations in Piezo1 that have been identified in patients with the RBC disease xerocytosis, also called dehydrated hereditary stomatocytosis (DHS) (*Zarychanski et al., 2012*; *Albuisson et al., 2013*; *Andolfo et al., 2013*;

**eLife digest** Within our bodies, cells and tissues are constantly being pushed and pulled by their surrounding environment. These mechanical forces are then transformed into electrical or chemical signals by cells. This process is crucial for many biological structures, such as blood vessels, to develop correctly, and is also a key part of our senses of touch and hearing.

In 2010, researchers discovered a group of ion channels—proteins embedded in the membrane that surrounds a cell—that open up when a force is applied and allow calcium and other ions to enter the cell. This movement of ions generates the electrical response of the cell to the applied force. However, not much is known about the roles of these 'Piezo' ion channels.

Red blood cells experience significant forces when they pass through narrow blood vessels. In a disease called xerocytosis, the red blood cells become severely dehydrated and shrink. In 2013, researchers discovered that patients with this disease have mutations in the gene that codes for the Piezo1 protein: a Piezo protein that has also been linked to a role in blood vessel development in embryos. This suggested that Piezo1 may regulate the volume of red blood cells.

Cahalan, Lukacs et al.—including some of the researchers who worked on the 2010 and 2013 studies—have now investigated the role of Piezo1 in red blood cells in more detail. Applying strong forces to red blood cells from mice caused calcium to rapidly enter cells through Piezo1 channels.

Cahalan, Lukacs et al. then deleted the Piezo1 gene from red blood cells. This made the cells larger and more fragile than normal cells because they contained too much water. To investigate how Piezo1 regulates water content, the cells were treated with a chemical compound called Yoda1. This compound was shown in a separate study by Syeda et al. to activate Piezo1 channels. Activating Piezo1 caused a second type of ion channel to open up as well, which allowed potassium ions and water molecules to leave the cell. This resulted in the cell becoming dehydrated.

This work raises the possibility that Piezo proteins are involved in other diseases where red blood cell volume is altered. In particular, many believe that Piezo1 may be involved in sickle cell disease, a possibility that can now be tested using the tools described in this study.

*Bae et al., 2013*). In addition, whole body treatment of zebrafish with Piezo1 morpholino affects RBC volume homeostasis (*Faucherre et al., 2014*). Finally, the Piezo1 locus has also been implicated in a genome-wide association screen for affecting the RBC mean corpuscular hemoglobin concentration (MCHC) in humans (*van der Harst et al., 2012*). However, whether any of these effects are due to Piezo1 expression on the RBCs themselves is not understood nor is the normal role of Piezo1 in mammalian RBC physiology.

## Results

We first investigated whether Piezo1 is expressed on mouse RBCs using $Piezo1^{P1-tdTomato}$ mice that express a Piezo1-tdTomato fusion protein from the $Piezo1$ locus (*Ranade et al., 2014*). Both peripheral blood RBCs (*Figure 1A*) and developing bone marrow pro-RBCs (*Figure 1B*) from $Piezo1^{P1-tdTomato}$ mice exhibited increased tdTomato fluorescence by flow cytometry compared to those from $Piezo1^{+/+}$ mice. Peripheral RBCs from $Piezo1^{P1-tdTomato}$ mice had clear expression of a ~320 kDa Piezo1-tdTomato fusion protein by Western blot (*Figure 1A*). To further investigate the role of Piezo1 in RBC physiology, we set out to genetically ablate it. Mice deficient in Piezo1 die in utero, so we deleted Piezo1 specifically in the hematopoietic system. We bred Vav1-iCre mice, which express Cre recombinase early in hematopoiesis (*Shimshek et al., 2002*), to mice where exons 20–23 of $Piezo1$ are flanked by loxP sites (P1$^f$), thus generating viable, fertile Vav1-iCre P1$^{f/f}$ (Vav1-P1cKO) mice (*Figure 1—figure supplement 1A*). Vav1-P1cKO lymphocytes exhibited >95% deletion of $piezo1$ transcript, demonstrating efficient Cre-mediated excision (*Figure 1—figure supplement 1C*). Hematological analysis of blood from Vav1-P1cKO mice revealed significant changes in RBC physiology without significant anemia (*Table 1*). Notably, compared to WT mice, Vav1-P1cKO mice had elevated (% of WT $\pm$ SEM) mean corpuscular volume (MCV, 109.51 $\pm$ 1.51) and mean corpuscular hemoglobin (MCH, 103.14 $\pm$ 0.48) and reduced mean corpuscular hemoglobin concentration (MCHC, 94.37 $\pm$ 1.08), suggesting that Piezo1-deficient RBCs were overhydrated. Since increased MCV can also be observed in the dehydrated RBCs in xerocytosis, we further tested whether Piezo1-deficient RBCs

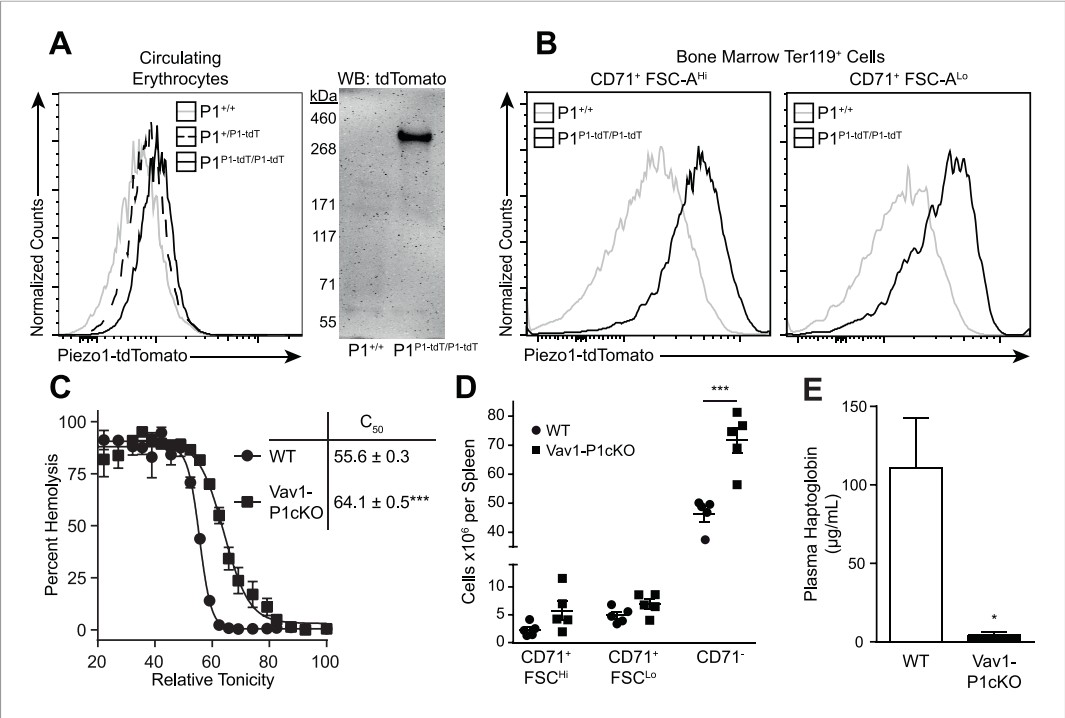

**Figure 1**. Deletion of Piezo1 in blood cells causes RBC fragility and splenic sequestration. (**A**) Left: flow cytometry histograms of tdTomato fluorescence on Ter-119+ peripheral blood red blood cells (RBCs). Rightward shifts indicate increased fluorescence. Right: Western blot for tdTomato from lysates of packed RBCs. (**B**) Flow cytometry histograms of tdTomato fluorescence from less (CD71+ FSC-A^Hi) and more (CD71+ FSC-A^Lo) mature RBC progenitor cells. (**C**) Percent hemolysis of blood of WT and Vav1-P1cKO mice when exposed to hypotonic solutions of indicated relative tonicity. $C_{50}$ values (relative tonicity at half maximal lysis) were calculated by fitting the data to a 4-parameter logistic sigmoidal curve. (**D**) Total number of Ter-119+ erythroid cells found in the spleens of WT and Vav1-P1cKO mice. (**E**) Plasma haptoglobin concentrations of both WT and Vav1-P1cKO mice as determined by ELISA. **A** and **B** are representative histograms and blots from three individual mice per genotype. Graphs in **C** and **E** result from individual experiments consisting of at least 3 WT and 3 Vav1-P1cKO mice each, with each experiment repeated 2, 3, and 3 times for **C**, **D**, and **E**, respectively. *$p < 0.05$, ***$p < 0.001$ by unpaired Student's $t$-test.

The following figure supplements are available for figure 1:

**Figure supplement 1**. Generation and validation of Vav1-P1cKO mice.

**Figure supplement 2**. Histology of WT and Vav1-P1cKO spleens.

were actually overhydrated. Overhydrated RBCs exhibit increased osmotic fragility and increased size as measured by forward scatter using flow cytometry. Blood from Vav1-P1cKO mice exhibited both of these characteristics (*Figure 1C* and *Figure 1—figure supplement 2A*), demonstrating that Piezo1-deficient RBCs are overhydrated. While Vav1-P1cKO RBCs were overhydrated, scanning electron microscopy of WT and Vav1-P1cKO RBCs revealed that Vav1-P1cKO RBCs had relatively normal discoid morphology, unlike more severe overhydration pathologies such as spherocytosis (*Figure 1—figure supplement 2B*). Regardless, these results suggest that Piezo1 expression on RBCs is a negative regulator of RBC volume.

Because changes in RBC volume commonly result in pathology in the spleen, we compared Vav1-P1cKO spleens with those of WT littermates. Although they appeared visibly darker and redder following H&E staining, spleens from Vav1-P1cKO mice exhibited normal formation of both red and white pulp without an evident expansion of red pulp or increased iron deposition (*Figure 1—figure supplement 2C,D*). However, flow cytometric analysis of splenic RBC subpopulations revealed an increased number of fully mature Ter119+ CD71− RBCs, but not immature Ter119+ CD71+ RBCs (*Figure 1D*), suggesting that the darker splenic color is due in part to retention of overhydrated

**Table 1.** Hematological indices from blood isolated from 8- to10-week-old WT and Vav1-P1cKO mice

| | WT ± SEM (n = 19) | Vav1-P1cKO ± SEM (n = 18) |
|---|---|---|
| RBC | 100 ± 0.58 | 96.60 ± 1.10* |
| HGB | 100 ± 0.54 | 99.50 ± 1.10 |
| HCT | 100 ± 0.51 | 105.59 ± 1.13*** |
| MCV | 100 ± 0.23 | 109.51 ± 1.51*** |
| MCH | 100 ± 0.25 | 103.14 ± 0.48*** |
| MCHC | 100 ± 0.26 | 94.37 ± 1.08*** |
| RDW | 100 ± 0.92 | 114.49 ± 2.64*** |

Data are pooled from four individual experiments, each experiment consisting of at least three age- and sex-matched mice per genotype. RBC: red blood cell count per unit volume; HGB: hemoglobin content; HCT: hematocrit; MCV: mean corpuscular volume; MCH: mean corpuscular hemoglobin; MCHC: mean corpuscular hemoglobin concentration; RDW: red cell distribution width. Indices for each mouse were normalized to the average value of WT mice from the same experiment. Statistics were calculated by two-tailed Mann–Whitney test. *p < 0.05, ***p < 0.001.

circulating mature RBCs. Consistent with this, immature splenic RBCs had similar forward scatter in WT and Vav1-P1cKO mice, indicating that they were of similar size, while fully mature RBCs in Vav1-P1cKO exhibited increased forward scatter indicative of increased size (*Figure 1—figure supplement 2A*). We also found that Vav1-P1cKO mice exhibited significantly lower plasma haptoglobin concentrations, indicative of intravascular hemolysis in vivo (*Figure 1E*). Thus, Piezo1-deficient RBCs have increased fragility and are aberrantly retained within the spleen, suggesting that Piezo1 helps maintain RBC integrity and normal recirculation.

Piezo1 is a mechanically activated, calcium-permeable non-selective cation channel. RBCs experience significant mechanical forces during circulation; we, therefore, sought to determine whether acute application of mechanical force could cause $Ca^{2+}$ influx and whether any $Ca^{2+}$ influx observed was dependent on Piezo1. However, many of the existing methods of mechanical stimulation available to us proved unsuitable for this purpose. Patch-clamp experiments proved impractical as conditions that allowed the formation of gigaohm seals in a cell-attached setting resulted in cell rupture upon application of negative pressure. Additionally, RBCs were also too small and fragile for indentation using a glass probe with or without simultaneous electrophysiological recording. Calcium imaging studies using shear stress in laminar flow chambers did not yield detectable increases in intracellular $Ca^{2+}$.

We, therefore, developed a mechanical stimulation assay combining the advantage of calcium imaging (no gigaohm seal necessary) and patch clamp (a pipette to capture the cell). We used micropipettes with a long, tapered tip and optimized their size and shape (tip diameter ~1–1.5 μm) so that RBCs could only partially enter into the pipette (*Figure 2A* and *Figure 2—figure supplement 1*). In the presence of 0.05% BSA, mechanical stimulation of these cells was possible without causing cell rupture by applying negative pipette pressure using a high-speed pressure clamp device (*Figure 2A*). This method allowed for quantitative and reproducible application of force while detecting changes in intracellular $Ca^{2+}$. Application of negative pressure induced a rapid rise in intracellular $Ca^{2+}$ levels in Fluo-4-loaded WT RBCs with a threshold of ~ −5 mmHg and a $P_{50}$ of −9.72 ± 0.51 mmHg (*Figure 2B,C* and *Video 1*). As seen in *Figure 2D*, repeated applications of negative pressure induced a mild rundown in the calcium response. To account for this, we normalized all responses in *Figure 2C* to the average of two maximal (−25 mmHg) stimuli flanking each test pulse (*Figure 2C*). Following removal of a half-maximal −10 mmHg stimulus, $Ca^{2+}$ levels declined to baseline with an average $t_{1/2}$ of 21 ± 1.8 s. This increase in intracellular $Ca^{2+}$ concentration must be a result of influx rather than receptor-mediated store release as RBCs do not possess intracellular stores. To conclusively determine whether this $Ca^{2+}$ influx was Piezo1-mediated, we subjected Vav1-P1cKO RBCs to similar negative pressure stimulation. We found that $Ca^{2+}$ influx was not detectable in any Vav1-P1cKO RBCs, even at pressures up to −35 mmHg (*Figure 2D*). Experiments using Vav1-P1cKO RBCs were conducted using the same pipettes that previously elicited $Ca^{2+}$ influx in WT RBCs, allowing for equivalent mechanical stimulation between genotypes. These results demonstrate that RBCs can respond to mechanical force by allowing $Ca^{2+}$ to enter the cell and that this $Ca^{2+}$ influx is dependent on Piezo1.

The ability of mechanical force to cause $Ca^{2+}$ entry through Piezo1, coupled with the observed overhydration of Piezo1-deficient RBCs, suggests an important role for Piezo1-dependent $Ca^{2+}$ influx in regulating RBC volume following mechanical stress. To directly test the effect that Piezo1 activation could have on RBC volume, we utilized the recently identified Piezo1-selective activator Yoda1 (*Syeda et al., 2015*). WT or Vav1-P1cKO RBCs were loaded with the calcium-sensitive dye Fluo-4 and

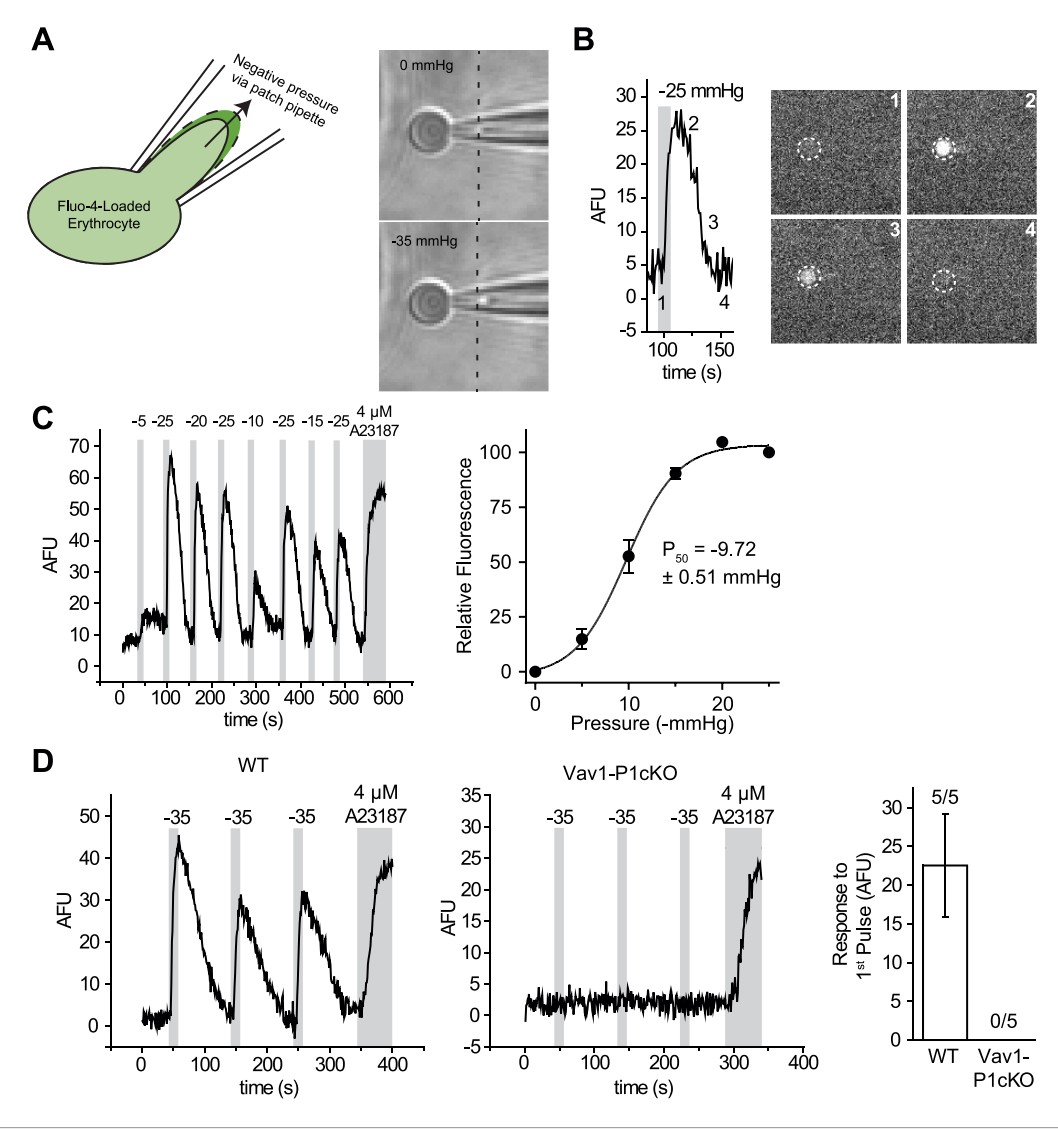

**Figure 2**. RBCs exhibit Piezo1-dependent Ca$^{2+}$ influx in response to mechanical stretch. (**A**) Left: cartoon representation of mechanical stretching of RBCs. Right: brightfield images of an individual RBC before (top) and during (bottom) application of −35 mmHg. Dotted line indicates starting location of RBC membrane prior to stretching. (**B**) Left: representative plot of background subtracted Fluo-4 fluorescence of an individual RBC following application of −25 mmHg for the time indicated by the gray shaded area. Right: images of the RBC plotted on left at the times indicated. (**C**) Left: representative plot of background subtracted Fluo-4 fluorescence from an individual RBC when subjected to pressure pulses of different magnitudes. Duration of the pulses is as indicated by shaded areas on plot; magnitude of pressure in mmHg is indicated above lines. Right: pressure-response curve (mean ± SEM) generated from 8 RBCs subjected to varying pressures. Responses to each different pressure were normalized to the average of the flanking −25 mmHg pulses, and the order of each different pressure applied was randomized for each separate RBC. (**D**) Representative plots from individual WT (Left) and Vav1-P1cKO (Right) RBCs subjected to pressure pulses as indicated in mmHg. Right graph represents mean ± SEM of fluorescence change in response to first −35 mmHg pulse from 5 WT and 5 Vav1-P1cKO RBCs subjected to mechanical stretching protocol as shown in plots. Numbers above graph indicate number of cells that had responses over 10 AFU out of total cells tested.

The following figure supplement is available for figure 2:

**Figure supplement 1**. Bright field image of representative pipette used to elicit Ca$^{2+}$ influx in RBCs.

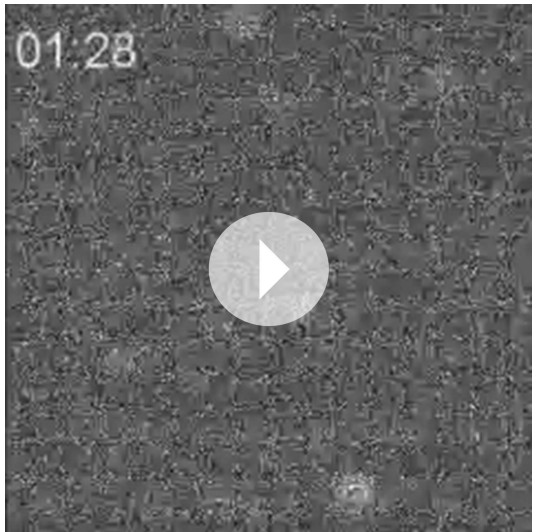

**Video 1.** Video of $Ca^{2+}$ influx into a RBC following mechanical stimulation. Time lapse video of a single Fluo-4-loaded RBC subjected to repeated 10-s pulses of −35 mmHg as indicated. Time scale is in min:s.

then treated with 15 μM Yoda1. Yoda1 caused robust increases in fluorescence of WT, but not in Vav1-P1cKO RBCs. In contrast, both WT and Vav1-P1cKO RBCs exhibited similar $Ca^{2+}$ influx in response to the $Ca^{2+}$ ionophore A23187 (*Figure 3A*).

A rise in intracellular $Ca^{2+}$ in response to Yoda1 would be expected to activate KCa3.1, also known as the Gardos channel, which can mediate $K^+$ efflux, $H_2O$ efflux, and RBC dehydration (*Maher and Kuchel, 2003*). In fact, RBCs from one strain of mice lacking KCa3.1 exhibit increased size and osmotic fragility similar to what is seen in Vav1-P1cKO mice (*Grgic et al., 2009*). We tested whether Yoda1 could cause RBC dehydration, and whether any possible dehydration was mediated through KCa3.1 activation by Piezo1-dependent $Ca^{2+}$ influx. Treatment of WT RBCs, but not Vav1-P1cKO RBCs, with 15 μM Yoda1 led to a rapid change in their shape, progressing from discocytes to echinocytes to spherocytes, similar to what has been demonstrated for treatment with A23187 (*Steffen et al., 2011*) (*Figure 3B*).

We further tested the osmotic fragility of blood following treatment with Yoda1 and/or the KCa3.1 antagonist TRAM-34 (*Wulff et al., 2000*). Incubation of WT, but not Vav1-P1cKO blood, for 30 min with 5 μM Yoda1 caused a marked reduction in RBC osmotic fragility that was prevented by a 10-min pretreatment with 2 μM TRAM-34 (*Figure 3C*). Both WT and Vav1-P1cKO RBCs exhibited reduced osmotic fragility when treated for 30 min with 1 μM A23187, demonstrating normal functionality of KCa3.1 in Vav1-P1cKO RBCs (*Figure 3D*). The lack of any $Ca^{2+}$ influx, shape changes, or changes in osmotic fragility of Vav1-P1cKO RBCs in response to Yoda1 further demonstrates the specificity of Yoda1 on Piezo1. Importantly, TRAM-34 did not block Piezo1 HEK293T cells transfected with Piezo1 (*Figure 3—figure supplement 1*). These data as a whole suggest a model shown in *Figure 3E* whereby activation of Piezo1 on RBCs leads to calcium influx, potassium efflux through KCa3.1 that is accompanied by water loss, resulting in RBC dehydration.

## Discussion

Many of the effects exerted by physical forces on cellular physiology remain unclear. Here, we have shown that these forces have a significant effect on RBC physiology by activating the mechanosensitive ion channel Piezo1. Our findings suggest a link between mechanical forces and RBC volume via $Ca^{2+}$ influx through Piezo1. The ability of RBCs to reduce their volume in response to mechanical forces could improve their ability to traverse through small-diameter capillaries and splenic sinusoids. Additionally, it is possible that this reduction in volume could aid in oxygen/$CO_2$ exchange in the periphery by concentrating hemoglobin within RBCs, which may promote release of oxygen from hemoglobin; in fact, mechanical stimulation of RBCs via optical tweezers has been shown to cause such a release of oxygen (*Rao et al., 2009*).

It was not previously clear whether the cause of RBC dehydration in DHS patients is due to direct or indirect mechanisms. We have demonstrated that genetic deletion of Piezo1 in blood cells leads to overhydrated, fragile RBCs. We have also shown that mechanical force can cause calcium influx into RBCs that is dependent on Piezo1 expression. We cannot exclude the possibility that deletion of Piezo1 alters RBC membrane properties, resulting in decreased activity of a separate mechanosensitive ion channel rather than calcium entering the cell through Piezo1 itself. We find this unlikely given the relatively mild overhydration of Vav1-P1cKO RBCs and the complete absence of any mechanically induced calcium influx in these cells. Finally, using the first identified selective small molecule activator of Piezo1, we have shown that calcium influx through Piezo1 dehydrates RBCs via the actions of KCa3.1. These findings are consistent with the observation of dehydrated RBCs from DHS patients

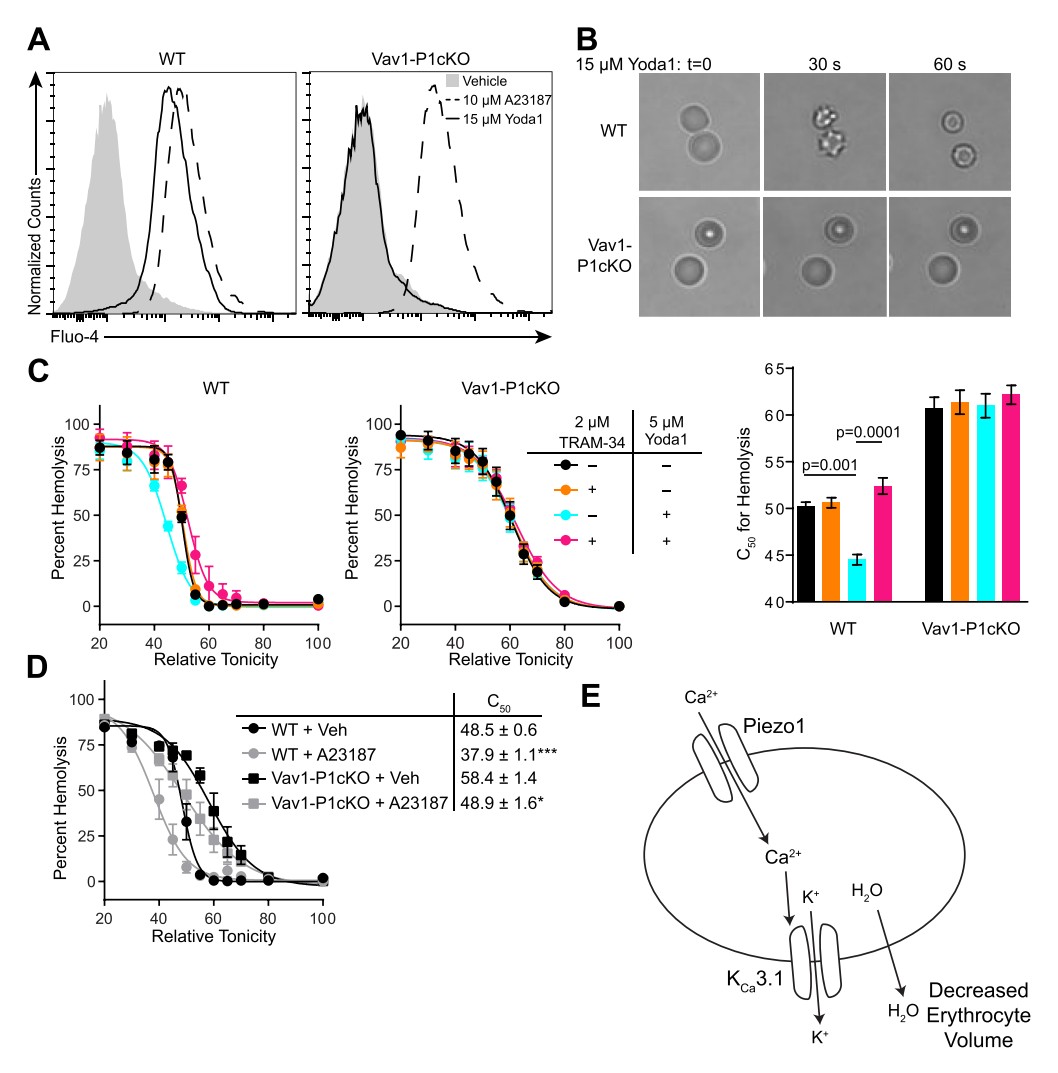

**Figure 3**. Piezo1 activation causes $Ca^{2+}$ influx and KCa3.1-dependent RBC dehydration. (**A**) Flow cytometry histograms of Fluo-4 fluorescence of WT (left) or Vav1-P1cKO (right) RBCs treated with vehicle (gray shaded), 15 μM Yoda1 (solid line), or 10 μM A23187 (dashed line) for 1 min as indicated. Rightward shifts in fluorescence indicate increased intracellular $Ca^{2+}$. (**B**) Brightfield images from RBCs from WT (top) or Vav1-P1cKO (bottom) RBCs at the indicated times after superfusion with 15 μM Yoda1. (**C**) Osmotic fragility (±SEM, n = 3) of blood from WT (left) or Vav1-P1cKO (middle) treated with 2 μM the KCa3.1 antagonist TRAM-34 and/or 5 μM Yoda1 as indicated. Blood was incubated with TRAM-34 or vehicle for 10 min, and then incubated with Yoda1 or vehicle for 30 min. Graph on right depicts $C_{50} ±$ SEM for hemolysis for the genotypes and treatments in the left graphs. p values were calculated using one-way ANOVA. (**D**) Osmotic fragility (±SEM, n = 3) of WT and Vav1-P1cKO blood treated with 1 μM of the $Ca^{2+}$ ionophore A23187 for 30 min. *p < 0.05; ***p < 0.001 compared to genotype-matched, vehicle-treated blood by Student's *t*-test. (**E**) Working model for how Piezo1 activation regulates RBC volume. Experiments were repeated the following number of times: **A**: 3, **B**: 3, **C**: 2, **D**: 2, with results from an individual experiment being presented.
The following figure supplement is available for figure 3:

**Figure supplement 1**. Tram-34 does not block Piezo1.

with gain-of-function Piezo1 mutations (*Zarychanski et al., 2012*; *Albuisson et al., 2013*; *Andolfo et al., 2013*; *Bae et al., 2013*), further supporting that the dehydration seen in DHS patients is due to increased Piezo1-mediated calcium influx in response to mechanical forces in RBCs themselves.

This work also demonstrates the power of combining both genetic and chemical approaches. Experiments performed on Piezo1-deficient background demonstrate the specificity of Yoda1; meanwhile, the acute effect of Yoda1 on RBC dehydration allows us to conclude that Piezo1 activity can regulate cell volume regulation via KCa3.1 activation and suggests that changes in RBC volume in Vav1-P1cKO mice are not a consequence of developmental compensation or non-cell autonomous mechanisms.

Piezo1 has been proposed by many to mediate a non-selective current observed in sickle cell disease (SCD) called $P_{Sickle}$, which acts upstream of KCa3.1 to mediate RBC sickling (*Lew et al., 1997*; *Ranney, 1997*; *Ma et al., 2012*; *Demolombe et al., 2013*). While KCa3.1 inhibitors can improve many hematological indices of SCD patients, they have thus far been ineffective in preventing painful vasculo-occlusive crises (*Ataga et al., 2008*). Generation of mice containing conditional deletion of Piezo1 in blood cells combined with full replacement of normal murine hemoglobin genes with sickling human hemoglobin genes could help determine whether Piezo1 mediates both $P_{Sickle}$ and the pathology of SCD. Inhibiting Piezo1 may modulate other pathways in addition to potentially blocking KCa3.1-dependent dehydration that may have therapeutic benefits in patients with SCD.

## Materials and methods

### Mice

Piezo1$^{P1-tdTomato}$ fusion knockin mice were generated as described in *Ranade et al. (2014)*. P1$^f$ mice were generated using the Piezo1$^{tm1a(KOMP)Wtsi}$ Knockout First, promoter driven targeting construct from KOMP. This construct was electroporated into Bruce4 (C57Bl/6-derived) ES cells, yielding homologously recombined ES cells that were injected into B6(Cg)-Tyr<c-2J>/J blastocysts at the TSRI Mouse Genetics Core. Chimeric mice from these injections were bred to C57Bl/6J mice, yielding germ line-transmitted mice. These mice were then bred to FLP-expressing mice to excise the neomycin resistance cassette, then once more to C57Bl/6J mice to remove FLP, yielding the P1$^f$ locus. Vav1-iCre mice on C57Bl/6J background (Stock # 008610) were purchased from the Jackson Laboratory. Mice were genotyped using the following primers: P1 F: CTT GAC CTG TCC CCT TCC CCA TCA AG, P1 WT/fl R: CAG TCA CTG CTC TTA ACC ATT GAG CCA TCT C, P1 KO R: AGG TTG CAG GGT GGC ATG GCT CTT TTT using Phire II polymerase (Thermo Scientific, Waltham, MA) with the following cycling conditions: Initial denaturation 98°C for 30 s, followed by 31 cycles of 98°C for 5 s, 65°C for 5 s, 72°C for 5 s, followed by a final hold of 72°C for 2 min. Vav1-iCre was genotyped using protocols as described by Jackson Laboratory. Reactions were separated on 2% agarose gels yielding the following band sizes: P1$^+$: 160 bp, P1$^f$: 330 bp, P1$^-$: 230 bp. It was noted that an obvious P1$^-$ band was found in most, but not all, Vav1-iCre$^+$ P1$^{f/f}$ mice. The rare mice that were genotyped as Vav1-iCre$^+$ P1$^{f/f}$ but did not exhibit a P1$^-$ band were found to have no changes in hematological indices and displayed Ca$^{2+}$ influx in response to Yoda1. Such mice were excluded from analysis, as they were likely a result of inefficient Cre-mediated deletion of Piezo1. All mice were housed in 12-hr light/dark cycle room with food and water provided ad libitum. All animal procedures were approved by the TSRI Institutional Animal Care and Use Committee.

### Hematology and microscopy

Blood was isolated from mice from either the retro-orbital plexus of mice anesthetized with isoflurane or from the heart of euthanized mice. Hematological data were collected using either a CellDyn 3700 (Abbott Laboratories, Abbott Park, IL) or a Procyte Dx (IDEXX Laboratories, Westbrook, ME) hematology analyzer.

Spleens from WT or Vav1-P1cKO were excised and fixed in 10% neutral buffer formalin for at least 24 hr, then processed and embedded in paraffin. 6-μm thick sections were cut, deparaffinized, and then were either stained with hematoxylin and eosin or with Prussian blue followed by Nuclear Fast Red. Red pulp and Prussian blue-stained areas were determined using Nikon Elements and were normalized to spleen area.

For scanning electron micrographs, blood obtained from cardiac puncture was resuspended in 149 mM NaCl + 2 mM HEPES pH 7.4 and centrifuged at 500×$g$ for 5 min. Cells were then resuspended in ice-cold fixative consisting of 2.5% glutaraldehyde in 0.1 M cacodylate buffer with 1 mM calcium. Aliquots of the fixed cells were allowed to settle on 12-mm coverslips previously coated with polylysine. The coverslips with adherent cells were subsequently washed in 0.1 M cacodylate buffer and postfixed in buffered 1% osmium tetroxide for 1 hr. The cells were washed extensively in distilled

water, and then gradually dehydrated with addition of ethanol to the water. The coverslips were critical point dried (tousimis Autosamdri 815) and the mounted onto SEM stubs with carbon tape. The stubs with attached coverslips were then sputter coated with 6 nm iridium (EMS model 150T S) for subsequent examination and documentation on a Hitachi S-4800 SEM (Hitachi High Technologies America Inc., Pleasanton CA) operating at 5 kV.

## Flow cytometry, Western blot, and ELISA

For analysis of Piezo1-tdTomato expression and splenic RBC composition, single cell suspensions of blood, bone marrow, or spleen were obtained in PBS containing 2% FBS. Cells were stained with fluorescently labeled antibodies, washed once, and data were acquired on a LSR-II flow cytometer (BD) and analyzed using FlowJo (FlowJo Inc, Ashland, OR). The following antibodies were used for staining, all at 1:100 dilution: PE-Cy7 Ter-119 (eBioscience, San Diego, CA), APC CD71 (eBioscience), APC-Cy7 CD45.2 (Biolegend, San Diego, CA). Absolute cell counts were determined using CountBright Absolute Counting Beads (Life Technologies, Carlsbad, CA) according to manufacturer's instructions. For analysis of $Ca^{2+}$ influx by flow cytometry, cells were loaded with Fluo-4 (Life Technologies) for 30 min at 37°C in PBS containing 0.05% BSA, washed once, and resuspended in the same buffer containing 2 mM $CaCl_2$. Compounds were added from DMSO stock at 500×, quickly vortexed, and incubated at room temperature for 2 min prior to acquisition.

For Western Blot of P1-tdTomato from erythrocytes, 20 µl of packed erythrocytes was lysed in ice-cold RIPA buffer (G-Biosciences, St. Louis, MO) containing 1:100 Protease Inhibitor cocktail (Cell Signaling Technologies, Danvers, MA). Protein amounts were calculated by Pierce micro BCA assay (Life Technologies), and then 20 µg of protein was loaded into 3–8% Novex Tris-Acetate polyacrylamide gels (Life Technologies) under denaturing conditions. Protein was then transferred to PVDF membranes using an iBlot transfer system (Life Technologies). Blots were incubated for 1 hr with 5% (wt/vol) milk in TBST at room temperature and then incubated with a rat anti-mCherry antibody (a gift from the laboratory of Hugh Rosen) at a concentration of 1 µg/ml in 5% milk/TBST overnight at 4°C. Blots were then incubated with HRP-conjugated goat anti-rat (Jackson Immunoresearch, West Grove, PA) for 1 hr at room temperature at a concentration of 1:10,000, and chemiluminescence was generated using Pierce ECL Plus (Life Technologies) reagent. Chemiluminescence was detected using a FluorChem Q (ProteinSimple, San Jose, CA).

For detecting plasma haptoglobin, plasma was isolated from blood by centrifugation at 2,000×g for 10 min at 4°C, and haptoglobin concentrations were calculated by ELISA (Genway Biotech, San Diego, CA) according to manufacturer's instructions.

## Compounds, osmotic fragility, and cell shape change

The identification and validation of Yoda1 is described (*Syeda et al., 2015*). A23187 and TRAM-34 were both purchased from Tocris (Bristol, United Kingdom) and were dissolved in DMSO.

To determine osmotic fragility, blood was first diluted in a ratio of 1:50 into normal saline (NS, 149 mM NaCl, 2 mM HEPES, pH 7.4), and then 10 µl of diluted blood was added to V-bottom 96-well plates. Solutions of varying tonicity were generated by mixing NS (100%) with 2 mM HEPES, pH 7.4 (0%). 250 µl of these solutions was added to the diluted blood and incubated for 5 min at room temperature. Plates were then spun down and 200 µl of the supernatant was transferred to flat-bottom 96-well plates. Absorbance at 540 nm was measured using either an Enspire (Perkin Elmer, Waltham, MA) or Cytation3 (BioTek, Winooski, VT) plate reader. $C_{50}$ values were determined by fitting the data to 4-parameter sigmoidal dose–response curves using Prism (Graphpad, La Jolla, CA). For examining the effects of compounds on osmotic fragility, blood was incubated with Yoda1 or A23187 for 30 min prior to addition into V-bottom 96-well plates; blood was incubated with TRAM-34 for 10 min prior to agonist addition when used. When treating with compounds, 2 mM $CaCl_2$ and 4 mM KCl were added to NS during incubation with compounds.

Cell shape change was visualized by allowing blood diluted in normal extracellular medium containing (in mM) 137 NaCl, 3.5 KCl, 2 $CaCl_2$, 1 $MgCl_2$, 10 HEPES, 10 dextrose with the addition of 0.05% BSA (NECM/BSA). Cells were allowed to settle onto uncoated glass coverslips. Non-adherent cells were washed away using whole-chamber perfusion, which was also used to deliver Yoda1 to the adherent erythrocytes. To monitor cell shape in response to this treatment, bright-field images were acquired every second for 2 min using a Axiovert S100 microscope (Zeiss, Oberkochen, Germany) at 40× magnification.

## Calcium imaging and mechanical stimulation of erythrocytes

Blood was diluted 1:1000 into NECM/BSA and incubated with 5 μM Fluo-4 AM (Life Technologies) while rotating for at least 1 hr at 4°C. Cells were then placed in an imaging chamber and washed via whole-chamber perfusion for removal of excess extracellular dye. Mechanical stimulation of erythrocytes was achieved by capturing individual RBCs in the ~1 μm aperture of custom-made micropipettes (as shown in *Figure 2*) and subsequently applying pulses of negative pressure to the pipette compartment using a High-Speed Pressure-Clamp (HSPC) device (ALA scientific, Farmingdale, NY). Micropipettes were pulled using 1.5/0.85-mm (OD/ID) borosilicate glass capillaries (Sutter Instruments, Novato, CA) with a P-97 Flaming/Brown micropipette puller (Sutter Instruments). The electrical resistance of such micropipettes varied in the range of 15–20 MΩ. Following expulsion of the erythrocyte after each measurement using positive pipette pressures, individual pipettes were reused for subsequent measurements (approx. 6–10 measurements possible with single pipettes before clogging), allowing for reliable comparison of the applied mechanical stimuli across experimental groups.

Fluorescent calcium measurements were performed using a Lambda DG4 fluorescent excitation source (Sutter Instruments) attached to a Zeiss Axiovert S100 microscope. Images were acquired at 1-s intervals using the Zen Pro acquisition suite (Zeiss). Background subtracted average fluorescence intensity was computed using FIJI (*Schindelin et al., 2012*) and plotted as arbitrary fluorescence units as a function of time.

## Statistics and data analysis

Data analysis and plots were generated using Prism (GraphPad). Osmotic fragility and pressure-response curves were generated by fitting the data to a 4-parameter sigmoidal dose–response curve. Student's *t*-test or one-way ANOVA was used to determine statistical significance where appropriate. Normal distribution of data was confirmed using the Shapiro–Wilk test.

## Acknowledgements

This work was supported by NIH NS083174 to AP, American Heart Association Grant 14POST20000016 to SMC, and a CIRM fellowship to SSR. AP is a Howard Hughes Medical Institute Investigator. We thank Jennifer Kefauver for assistance with Western blotting and Malcolm Robert Wood for assistance with scanning electron microscopy.

## Additional information

### Funding

| Funder | Grant reference | Author |
|---|---|---|
| National Institutes of Health (NIH) | NS083174 | Ardem Patapoutian |
| Howard Hughes Medical Institute (HHMI) | Investigator | Ardem Patapoutian |
| California Institute for Regenerative Medicine (CIRM) | | Sanjeev S Ranade |
| American Heart Association (AHA) | Postdoctoral Fellowship 14POST20000016 | Stuart M Cahalan |

The funders had no role in study design, data collection and interpretation, or the decision to submit the work for publication.

### Author contributions

SMC, VL, MB, Conception and design, Acquisition of data, Analysis and interpretation of data, Drafting or revising the article; SSR, Conception and design, Acquisition of data, Analysis and interpretation of data; SC, Analysis and interpretation of data, Drafting or revising the article; AP, Conception and design, Analysis and interpretation of data, Drafting or revising the article

### Ethics

Animal experimentation: All animal procedures were approved by the TSRI Institutional Animal Care and Use Committee (#08-0136).

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
