## [Decision Letter]

Thank you for sending your work entitled “Piezo1 links mechanical forces to red blood cell volume” for consideration at *eLife*. Your article has been favorably evaluated by Janet Rossant (Senior editor), a Reviewing editor, and three reviewers. Two of the reviewers, Eric Honoré and Jon Levine, have agreed to share their identity.

The Reviewing editor and the reviewers discussed their comments before we reached this decision, and the Reviewing editor has assembled the following comments to help you prepare a revised submission. We are including the three reviews (lightly edited) at the end of this letter, as there are some specific and useful suggestions in them that will not be repeated in the summary here.

All of the reviewers were impressed with the importance and novelty of your work and had no major concerns. We look forward to receiving the revised version of your manuscript and are delighted that you chose to send this important work to *eLife*.

Reviewer #1:

In the manuscript “Piezo1 links mechanical forces to red blood cell volume” the authors investigate the molecular mechanism by which Piezo1 ion channel activity contributes to the dehydrating red blood cell disease Xerocytosis. By using conditional Piezo1 deletion from red blood cells, a novel mechanical assay combining fluorescent calcium-imaging with cell membrane stretch and the novel chemical Piezo1-specific agonist described in the accompanying manuscript they demonstrate that calcium-influx is completely dependent on Piezo1 and leading to cell dehydration via the ion channel KCAa3.1.

The finding is novel because it clearly links activity of Piezo1 to Xerocytosis for which previous evidence coming from Piezo1 gain-of-function mutations, zebrafish Piezo1 morpholino treatment and genome-wide association studies was only indirect. It is significant because it identifies Piezo1 as a molecular target for Xerocytosis, demonstrates that chemical modulation of Piezo1 activity can directly red blood cell volume and opens the possibility to study mechanisms of sickle cell disease.

With one exception (see below) I find that the conclusions are well supported by the data and I anticipate my suggestions for experiments or careful discussion will address my concern. The manuscript writing is lacking some carefulness and below I am listing a number of minor points that should all be addressed.

Major concern:

1) Since Vav1-P1cKO cells are drastically different in shape I am not convinced that the mechanical stimulus applied in the calcium-imaging/membrane stretch experiments (Figure 2 and Figure 2) is equivalent to the control (wild-type) cells. Other mechanosensitive ion channels might be present in these cells and their activation not obvious due to changed cell elasticity. While the data on chemical Piezo1 activation by Obi1 (Figure 3) convincingly demonstrate that Piezo1 is able to induce dehydration, the conclusion that calcium-influx upon membrane-stretch is entirely dependent on Piezo1 is still overreaching. To address this issue experimentally, the authors could combine fluorescent calcium-imaging with cell-poking (no patch pipette). If cells that are briefly treated with Obi1 (e.g. 1minute exposure to Obi1 and then wash) are similar in elasticity I would expect an extension/activity profile similar to that obtained from untreated cells. In an alternative experiment the authors could transiently induce calcium influx into wild-type cells to induce dehydration and subsequently assay mechanical sensitivity of Piezo1. At the least this issue has to be discussed and Figure 3 has to be modified to show that chemical (and only perhaps mechanical) activation of Piezo1 leads to calcium influx and cell dehydration.

Minor comments:

1) In the Results, the authors report that RBC have ‘similar forward scatter in WT and Vav1-P1cKO mice’, but they do not give an explanation or interpretation of this result.

2) When explaining the novel mechanical stimulation assay the authors fail to mention that cells are loaded with fluorescent calcium indication (Fluo-4) to monitor calcium levels.

3) The sentence ‘WT or Vav1-P1cKO RBCs loaded with the calcium-sensitive dye Fluo-4 and then treated 15uMObi1’ does not make sense.

4) Figure 1: The panel is lacking axis labels. Please, add.

5) All experiments: please indicate the actual number of experiments (n) instead of general statements (e.g. at least 3 WT mice).

6) Table 1: Please indicate the meanings of all abbreviations. Also, it would be interesting and helpful to cite (or measure) these values from human Xerocytosis patients in order to put the values in perspective.

7) Figure 2: The time course of calcium-increase with respect to mechanical stimulation is not clear to me. Does it happen upon release of pressure? Please clarify this.

8) Figure 2: The sequence of amplitudes of mechanical stimuli is unusual and I would normally expect successively larger stimuli. What was the reason for this protocol and how will it influence channel response due to inactivation?

9) Figure 3: The panel is lacking axis labels. Please, add.

10) Figure 3 and Figure 3: Please add error bars to all data point in these panels.

Reviewer #2:

Patapoutian and colleagues have previously reported that Piezo1 gain-of-function mutations are associated with Xerocytosis (dehydration of erythrocytes). In the present manuscript, they report that Piezo1 plays a major role in regulating the volume of red blood cells (RBCs). Opening of Piezo1 in response to mechanical stress is responsible for a calcium influx in RBCs. Increasing intracellular calcium leads to the opening of the KCa3.1 Gardos channel, loss of K^+^ ions and dehydration. Deletion of Piezo1 specifically in hematopoietic cells caused RBC fragility and sequestration in the spleen. By contrast, Piezo1 opening with the newly discovered opener Obi1 (see companion paper), results in RBC dehydration. Importantly, this effect is absent when Piezo1 is deleted in RBCs. It is concluded that Piezo1 may contribute to Psickle.

This is a very important paper for understanding how RBCs regulate their volume and help to decipher the mechanisms involved in RBCs pathology, including Xerocytosis and Sickle Cell disease. Moreover, this paper further demonstrates “physiologically” the effect of the Piezo1 opener Obi1 (see companion paper). The manuscript is very well written and presented. The technical quality of the work appears to be sound. I don't see much to change in this report, which could be published in its present form. This manuscript is a very nice complement of the other report concerning the discovery of Obi1. I only have a few minor points and suggestions which may help to further improve the quality of this great paper.

Minor points and suggestions:

1) Is Obi1 active on the Piezo1 gain-of-function mutations causing Xerocytosis? Please, cite the work of Sachs and collaborators also describing gain-of-function mutations on Piezo1: Xerocytosis is caused by mutations that alter the kinetics of the mechanosensitive channel PIEZO1. Bae C, Gnanasambandam R, Nicolai C, Sachs F, Gottlieb PA. Proc Natl Acad Sci USA. 2013.

2) Please mention at the end of the Discussion the comments of Demolombe and colleagues already suggesting a role for Piezo1 in Pscickle: Slower Piezo1 inactivation in dehydrated hereditary stomatocytosis (xerocytosis). Demolombe S, Duprat F, Honoré E, Patel A Biophys J. 2013 Aug 20;105(4):833-4.

3) Please cite the work of Ellory and colleagues, demonstrating the possible involvement of SACs in Psickle: The conductance of red blood cells from sickle cell patients: ion selectivity and inhibitors. Ma YL, Rees DC, Gibson JS, Ellory JC. J Physiol. 2012 May 1;590(Pt 9):2095-105. doi: 10.1113/jphysiol.2012.229609. Epub 2012 Mar 12.

4) Please cite the Nilius-TINS 2012 review about Piezo1

Reviewer #3:

The Piezos are a novel pair of ion mechanically gated ion channels recently cloned by Patapoutian and colleagues. Others and they have shown a role of Piezo 1 in vascular biology. In the present study they have complemented these findings by showing a role of Piezo 1 in erythrocyte function. A well designed and executed study, I have but one very minor question they might be able to comment toward. While they do note that combining interventions affecting sickle cell hemoglobin and Piezo 1 might provide novel and important information toward novel treatment of sickle cell disease, I was wondering if there is anything known about the role of mechanical forces, including swelling and shrinking, on oxygen exchange, as the major force change they are studying would occur at the site of oxygen exchange. If not, a comment to this effect might be worth introducing in their manuscript.

---

## [Author Response]

Below are the reviews for the three referees, and our responses to their critiques. As mentioned in the accompanying paper, Obi1 is now renamed Yoda1.

Reviewer #1:

*Major concern*:

*1) Since Vav1-P1cKO cells are drastically different in shape I am not convinced that the mechanical stimulus applied in the calcium-imaging/membrane stretch experiments (*Figure 2
*and*
Figure 2*) is equivalent to the control (wild-type) cells*.

The shape of Piezo1-deficient RBCs are not drastically different compared to wild-type RBCs; in fact they are relatively normal, discoid-shaped RBCs. This is in contrast to many other RBC diseases such as spherocytosis or sickle cell disease. To further demonstrate this, we have included images of scanning electron micrographs of both WT and Vav1-P1cKO blood obtained under similar magnification. These images are provided as Figure 1—figure supplement 2.

*Other mechanosensitive ion channels might be present in these cells and their activation not obvious due to changed cell elasticity. While the data on chemical Piezo1 activation by Obi1 (*Figure 3*) convincingly demonstrate that Piezo1 is able to induce dehydration, the conclusion that calcium-influx upon membrane-stretch is entirely dependent on Piezo1 is still overreaching*.

The reviewer is indeed correct that we cannot exclude the potential for other mechanosensitive ion channels being responsible for the calcium influx at this point in time. We have now mentioned this possibility in the Discussion as follows.

We have further modified the sentence in our Abstract to now say “…and that this entry is dependent on Piezo1 expression”, which more accurately reflects the data.

*To address this issue experimentally, the authors could combine fluorescent calcium-imaging with cell-poking (no patch pipette). If cells that are briefly treated with Obi1 (e.g. 1minute exposure to Obi1 and then wash) are similar in elasticity I would expect an extension/activity profile similar to that obtained from untreated cells. In an alternative experiment the authors could transiently induce calcium influx into wild-type cells to induce dehydration and subsequently assay mechanical sensitivity of Piezo1*.

We tried, but find it technically very difficult to perform cell poking experiments on RBCs. RBCs are very small and either just deflect or roll away from the indentation apparatus. We have further clarified this point in the manuscript.

*At the least this issue has to be discussed and*
Figure 3
*has to be modified to show that chemical (and only perhaps mechanical) activation of Piezo1 leads to calcium influx and cell dehydration.*

We have eliminated the reference to “Mechanical Forces” in Figure 3 with regards to how activation of Piezo1 may cause cell dehydration.

*Minor comments*:

*1) In the Results, the authors report that RBC have ‘similar forward scatter in WT and Vav1-P1cKO mice’, but they do not give an explanation or interpretation of this result*.

We have clarified that forward scatter is a measure of size.

*2) When explaining the novel mechanical stimulation assay the authors fail to mention that cells are loaded with fluorescent calcium indication (Fluo-4) to monitor calcium levels*.

We have now mentioned that RBCs were loaded with Fluo-4.

*3) The sentence ‘WT or Vav1-P1cKO RBCs loaded with the calcium-sensitive dye Fluo-4 and then treated 15uMObi1’ does not make sense*.

We have revised this sentence.

*4)*
Figure 1*: The panel is lacking axis labels. Please, add*.

Figure 1 now has an axis label of “Normalized counts”.

*5) All experiments: please indicate the actual number of experiments (n) instead of general statements (e.g. at least 3 WT mice)*.

In all cases we have now indicated both the number of experiments and the number of mice per experiment where applicable.

*6)*
Table 1*: Please indicate the meanings of all abbreviations. Also, it would be interesting and helpful to cite (or measure) these values from human Xerocytosis patients in order to put the values in perspective*.

The meanings for all definitions have been added to Table 1. Additionally, we have further clarified in the text the effect that blood from xerocytosis patients can exhibit increased MCV despite being dehydrated.

*7)*
Figure 2*: The time course of calcium-increase with respect to mechanical stimulation is not clear to me. Does it happen upon release of pressure? Please clarify this*.

The response initiates upon application of pressure and declines following the release of the pressure. The brevity of the pressure pulses, as compared to the length of the full trace, may make this relationship difficult to appreciate, we have therefore shaded the times in Figure 2 at which pressure was applied for increased clarity.

*8)*
Figure 2*: The sequence of amplitudes of mechanical stimuli is unusual and I would normally expect successively larger stimuli*. *What was the reason for this protocol and how will it influence channel response due to inactivation?*

Such a protocol was needed due to the gradual rundown of signal over time with repeated mechanical stimuli. The order of pressure pulses prior to the -25 mmHg “normalizing” pulses was randomized for each cell tested. We have inserted a description of this into the text and into the figure legend for Figure 2.

*9)*
Figure 3*: The panel is lacking axis labels. Please, add*.

Figure 3 now has an axis label of “Normalized counts”.

*10)*
Figure 3
*and*
Figure 3*: Please add error bars to all data point in these panels*.

The experiments have been repeated with 3 biological replicates per genotype per experiment, and error bars are now shown as mean ± SEM. For these experiments, new batches of Yoda1 and A23187 were used, and it was found that 5 μM and 1 μM of the new batches of Yoda1 and A23187, respectively, yielded optimal dehydration of WT RBCs.

Reviewer #2:

*Minor points and suggestions*:

1) Is Obi1 active on the Piezo1 gain-of-function mutations causing Xerocytosis?

We now show that Yoda1 is indeed active on Piezo1 gain-of-function mutations associated with xerocytosis; in fact, gain-of-function mutants have increased sensitivity to Obi1 in FLIPR experiments. These results are now included in the accompanying Syeda et al. paper as Figure 1—figure supplement 1 and are shown below.

Author response image 1.**DOI:**
http://dx.doi.org/10.7554/eLife.07370.014

*Please, cite the work of Sachs and collaborators also describing gain-of-function mutations on Piezo1: Xerocytosis is caused by mutations that alter the kinetics of the mechanosensitive channel PIEZO1. Bae C, Gnanasambandam R, Nicolai C, Sachs F, Gottlieb PA. Proc Natl Acad Sci USA. 2013*.

The reference is cited.

*2) Please mention at the end of the Discussion the comments of Demolombe and colleagues already suggesting a role for Piezo1 in Pscickle: Slower Piezo1 inactivation in dehydrated hereditary stomatocytosis (xerocytosis). Demolombe S, Duprat F, Honoré E, Patel A Biophys J. 2013 Aug 20;105(4):833-4*.

*3) Please cite the work of Ellory and colleagues, demonstrating the possible involvement of SACs in Psickle: The conductance of red blood cells from sickle cell patients: ion selectivity and inhibitors. Ma YL, Rees DC, Gibson JS, Ellory JC. J Physiol. 2012 May 1;590(Pt 9):2095-105.* doi: 10.1113/jphysiol.2012.229609*. Epub 2012 Mar 12*.

4) Please cite the Nilius-TINS 2012 review about Piezo1

All of the above references are now cited

Reviewer #3:

*The Piezos are a novel pair of ion mechanically gated ion channels recently cloned by Patapoutian and colleagues. Others and they have shown a role of Piezo 1 in vascular biology. In the present study they have complemented these findings by showing a role of Piezo 1 in erythrocyte function. A well designed and executed study, I have but one very minor question they might be able to comment toward. While they do note that combining interventions affecting sickle cell hemoglobin and Piezo 1 might provide novel and important information toward novel treatment of sickle cell disease, I was wondering if there is anything known about the role of mechanical forces, including swelling and shrinking, on oxygen exchange, as the major force change they are studying would occur at the site of oxygen exchange. If not, a comment to this effect might be worth introducing in their manuscript*.

This is an interesting possibility. Thank you for this suggestion. There isn’t much literature regarding the role of mechanical forces in oxygen exchange. One paper has described that mechanical stretching of RBCs by optical tweezers can release oxygen from hemoglobin (now cited, Rao et al., Biophysical Journal, 2009), though the precise mechanism by which this happens is unknown. The involvement of Piezo1 in such a process is possible, and we will look into it in the future. For now, we have added the following sentence in Discussion section: “Additionally, it is possible that this reduction in volume could aid in oxygen/CO_2_ exchange in the periphery by concentrating hemoglobin within RBCs, which may promote release of oxygen from hemoglobin; in fact, mechanical stimulation of RBCs via optical tweezers has been shown to cause such a release of oxygen.”